# Impact of sleep on the microbiome of oral biofilms

**Maki Sotozono**[1], **Nanako Kuriki**[2], **Yoko Asahi** [2]*, **Yuichiro Noiri**[1], **Mikako Hayashi**[2], **Daisuke Motooka**[3], **Shota Nakamura**[3], **Mikiyo Yamaguchi**[2], **Tetsuya Iida**[3], **Shigeyuki Ebisu**[2]

**1** Division of Cariology, Operative Dentistry and Endodontics, Department of Oral Health Science, Niigata University Graduate School of Medical and Dental Sciences, Niigata, Japan, **2** Department of Restorative Dentistry and Endodontology, Osaka University Graduate School of Dentistry, Suita, Osaka, Japan, **3** Research Institute for Microbial Diseases, Osaka University, Suita, Osaka, Japan

\* yoko-a@dent.osaka-u.ac.jp

## Abstract

Dysbiosis of the oral microbiome is associated with diseases such as periodontitis and dental caries. Because the bacterial counts in saliva increase markedly during sleep, it is broadly accepted that the mouth should be cleaned before sleep to help prevent these diseases. However, this practice does not consider oral biofilms, including the dental biofilm. This study aimed to investigate sleep-related changes in the microbiome of oral biofilms by using 16S rRNA gene sequence analysis. Two experimental schedules—post-sleep and pre-sleep biofilm collection—were applied to 10 healthy subjects. Subjects had their teeth and oral mucosa professionally cleaned 7 days and 24 h before sample collection. Samples were collected from several locations in the oral cavity: the buccal mucosa, hard palate, tongue dorsum, gingival mucosa, tooth surface, and saliva. *Prevotella* and *Corynebacterium* had higher relative abundance on awakening than before sleep in all locations of the oral cavity, whereas fluctuations in *Rothia* levels differed depending on location. The microbiome in different locations in the oral cavity is affected by sleep, and changes in the microbiome composition depend on characteristics of the surfaces on which oral biofilms form.

## Introduction

The Human Microbiome Project (2007–2017) revealed that an enormous number of bacteria inhabit the human body, forming indigenous microbiomes in each habitat, including the gut, oral cavity, skin, vagina, and airways. In the Human Microbiome Project, 16S rRNA gene sequence analysis was performed to investigate the microbiomes of different parts of the body [1]. More than 700 species of bacteria are present in various locations in the oral cavity, including the tooth surface, tongue, soft and hard palates, gingival mucosa, and buccal mucosa [2, 3]. These oral biofilms have different microbiome compositions in different locations [4]. The oral microbiome is one of the most diverse human microbiomes. Studies have reported characteristics of the oral microbiome and intra- and interindividual variations; indeed, the microbial diversity in saliva and dental biofilms is widely different among individuals and is affected

**Data Availability Statement:** The 16S rRNA sequencing data is available from DDBJ under the accession number DRA011991.

**Funding:** This study was supported by JSPS KAKENHI under Grant #17H04384 (SE),

#20K23104 (MS) and SECOM Science and Technology Foundation (MH). The funders had no role in study design, data collection and analysis, decision to publish, or preparation of the manuscript.

**Competing interests:** The authors have declared that no competing interests exist.

by the behavior of the host, such as oral self-care [5]. Many other factors also influence the oral microbiome, including salivary enzymes, pH, host immunity, and antibacterial agents [6].

The dental biofilm is closely associated with dental caries and periodontitis, which are chronic infectious diseases worldwide [7]. Dental caries are related to low pH on the tooth surface caused by acid produced from carbohydrates by oral bacteria [8–10]. Periodontitis is strongly associated with bacteria including *Porphyromonas gingivalis*, *Treponema denticola*, and *Tannerella forsythia* [11]. An imbalance in the oral microbial community (i.e., dysbiosis) was found to be associated with oral infections [12, 13], just as some intestinal diseases are caused by gut dysbiosis. Acidogenic and acid-tolerant bacteria are involved in dental caries, while strictly anaerobic proteolytic and alkaliphilic bacteria are involved in periodontitis [14–17].

In addition to these diseases associated with the dental biofilm, halitosis has been associated with the tongue-coating biofilm and its metabolites in many studies [18]. The tongue-coating biofilm produces volatile sulfur compounds [18–20], such as methyl mercaptan, hydrogen sulfide, and dimethyl sulfide [21–24].

Salivary flow during sleep is decreased compared with the waking hours [25]. The number of bacteria in saliva is highest upon awakening because numbers increase rapidly during sleep [26]. It is generally thought that production of volatile sulfur compounds is highest in the morning [27] because of the increased bacterial number in saliva [26]. Therefore, it is usually recommended to perform oral self-care such as brushing teeth before sleeping, to help prevent oral disorders. In general, clinicians and patients consider that oral care before sleep is important. However, this idea is based only on the bacterial number in saliva and biofilm metabolites, and does not consider the role of the microbiome composition. Therefore, we previously investigated the effect of sleep on the characteristics of dental biofilms formed in an *in situ* model [28]. We found that the number of biofilm-forming bacteria did not change significantly before and after sleep, but genera associated with periodontitis (i.e., *Prevotella* and *Fusobacterium*) were relatively more abundant on awakening than during the day [28].

The salivary microbiome reportedly has circadian oscillation [29]; however, the relationship between oral biofilms and the circadian rhythm has not been clarified. Moreover, the effect of sleep on the microbiome of oral biofilms has not been sufficiently investigated because it is difficult to perform such experiments.

We hypothesized that the microbiome of oral biofilms is affected by sleep, as was true for the experimental dental biofilm. Therefore, in this study, we used 16S rRNA gene sequence analysis to investigate changes in the microbiome of biofilms in various oral locations before and after sleep to determine those changes associated with sleep.

## Materials and methods

### Selection of study subjects

Ten healthy volunteers (six men and four women, 27–32 years-of-age) were recruited from the students and staff of Osaka University Graduate School of Dentistry. We defined healthy subjects as previously reported [30]. Written informed consent was obtained from all subjects. No clinical signs of gingivitis, periodontitis, or caries were detected and no systemic disease was observed in any of the subjects. For each participant, we recorded the total number of decayed, missing, or filled teeth as an index of dental caries, and the Community Periodontal Index as an index of periodontal disease. Table 1 shows information on subject characteristics. Subjects abstained from antibiotics 3 months before this study. The study design was reviewed and approved by the Ethics Committee of the Osaka University Graduate School of Dentistry (H30-E42) and conducted according to the guidelines of the Declaration of Helsinki.

**Table 1. Characteristics of subjects.**

| Subject number | Sex | Age (years) | DMF | CPI |
|:---:|:---:|:---:|:---:|:---:|
| 1 | F | 31 | 14 | 0 |
| 2 | M | 31 | 0 | 0 |
| 3 | F | 29 | 12 | 0 |
| 4 | F | 31 | 2 | 0 |
| 5 | M | 33 | 3 | 0 |
| 6 | M | 28 | 1 | 0 |
| 7 | M | 28 | 4 | 0 |
| 8 | M | 33 | 7 | 0 |
| 9 | M | 32 | 0 | 0 |
| 10 | M | 29 | 2 | 0 |

F, female; M, male. Dental caries were quantified as the total number of teeth that were decayed, missing, or filled (DMF). The Community Periodontal Index (CPI) was used as an index of periodontal disease. No clinical signs of caries, gingivitis, or periodontitis were detected, and no systemic disease was observed in any of the subjects.

## Sample collection and DNA extraction

The experimental schedule is shown in Fig 1. In this study, all 10 subjects participated in two experimental schedules (post- and pre-sleep sample collection) with a minimum of 2 weeks between the two schedules. In both schedules, subjects had their teeth and oral mucosa cleaned twice by a specialist, the first cleaning 7 days before sample collection and the second cleaning 24 h before sample collection. Biofilm samples were collected at 08:00 in the post-sleep schedule, and at 00:00 (midnight) in the pre-sleep schedule. The subjects avoided oral self-care for the 24 h between the second professional cleaning and sample collection. All subjects slept for 8 h, from 00:00 to 08:00.

Schematic of schedules and timing of sample collection. Subjects had their teeth and oral mucosa professionally cleaned 7 days before sample collection in both schedules. Subjects had

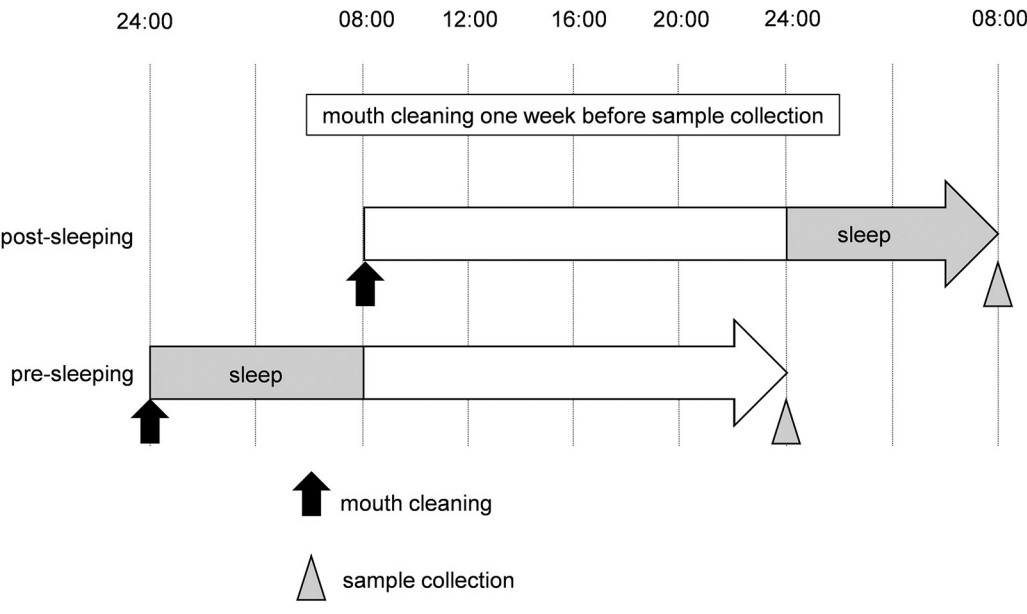

**Fig 1. Experimental schedules.**

their teeth and oral mucosa professionally cleaned again at 08:00 in the post-sleep schedule and at 00:00 (midnight) in the pre-sleep schedule (indicated by black arrows) and were instructed not to perform oral self-care after this cleaning. Biofilm samples were collected 24 h after this second professional cleaning (indicated by arrowheads). All subjects slept for 8 h (from 00:00 to 08:00) in both schedules.

Biofilm samples were collected in accordance with the methods of the Manual of Procedures for the Human Microbiome Project, with partial modification [1, 31]. In brief, biofilm samples were collected from the buccal mucosa, hard palate, tongue dorsum, and gingival mucosa with an Isohelix swab (Sci Trove, Kent, United Kingdom). Subgingival and supragingival dental biofilm samples were collected from the maxillary right first molar, mandibular left first molar, maxillary right central incisor, mandibular left central incisor, maxillary left first premolar, and mandibular right first premolar with Gracey curettes. Unstimulated saliva was collected. Collected biofilm and saliva samples were immediately processed for DNA extraction with a DNeasy® PowerSoil® DNA Isolation Kit (QIAGEN, Hilden, Germany).

### 16S rRNA sequence analysis

The V1–V2 region of bacterial 16S rRNA genes was amplified using the primer set 27F (`AGR GTT TGATCMTGG CTC AG` [32, 33]) and 338R (`TGC TGC CTC CCG TAG GAG T` [34]). The Illumina library was prepared by the tailed PCR method in accordance with the Illumina 16S Metagenomic Sequencing Library Preparation Guide. Sequencing (251-bp paired-end) was performed using MiSeq Reagent Kit v2 (500 cycles) and a MiSeq instrument (Illumina Inc.). The sequences were processed and clustered into operational taxonomic units (OTUs) with a 97% similarity cutoff by using the Greengenes database (v. 13.8) [35]. The results of sequences were analyzed by using the Quantitative Insights into Microbial Ecology pipeline (v. 1.9.1) [36].

The 16S rRNA amplicon sequencing data from this study was deposited in the DNA Data Bank of Japan (DDBJ) with accession number **DRA011991**.

### Statistical analysis

Non-metric multidimensional scaling (NMDS) and permutational multivariate analysis of variance (PERMANOVA) were performed with R software v. 3.6.1 (R Core Team, Vienna, Austria) and the vegan package. In PERMANOVA, $P < 0.05$ was considered a statistically significant difference between experimental schedules. The Wilcoxon signed rank test was used to evaluate alpha diversity and the relative abundance of each genus; $P < 0.05$ was considered a statistically significant difference between schedules. The Friedman test was used to evaluate beta diversity; $P < 0.05$ was considered a statistically significant difference between sites. Statistical analysis was performed and graphical outputs were prepared using IBM SPSS Statistics (v. 22.0, IBM SPSS Inc., Endicott, New York).

## Results

### Profiles of microbiome at each location in the oral cavity

The total number of reads was 10,169,418, and the average read count in samples was 72,639. Alpha diversity (Chao1 and Shannon indexes) are shown in Fig 2A, and beta diversity (UniFrac distances) in Fig 2B. There was a significant difference in the Chao1 index between the post-sleeping and pre-sleeping schedules at the buccal mucosa and gingival mucosa (buccal mucosa $P = 0.022$, gingival mucosa $P = 0.037$). There was also a significant difference in the Shannon index at the buccal mucosa ($P = 0.007$). No significant difference in intraindividual

(A)

### Chao 1 index

### Shannon index

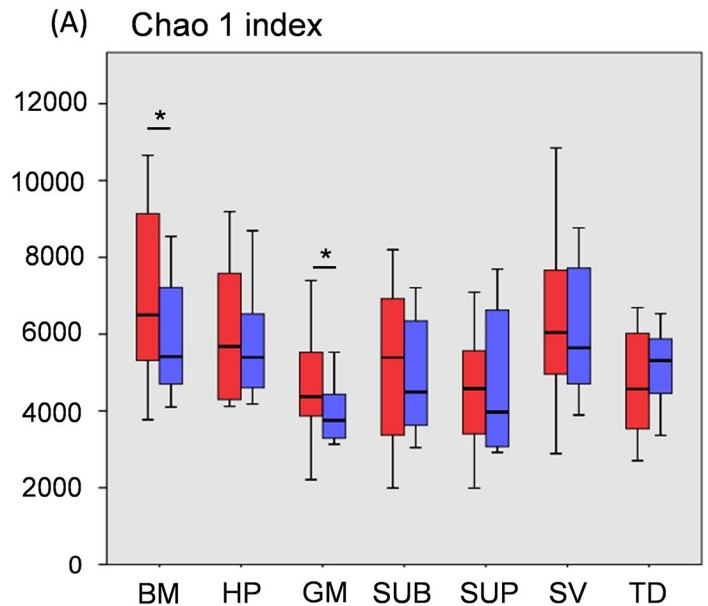
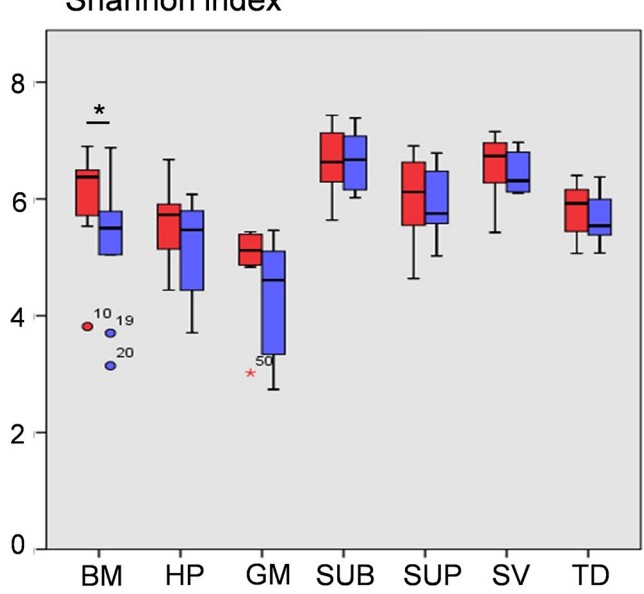

(B)

### unweighted UniFrac distance

### weighted UniFrac distance

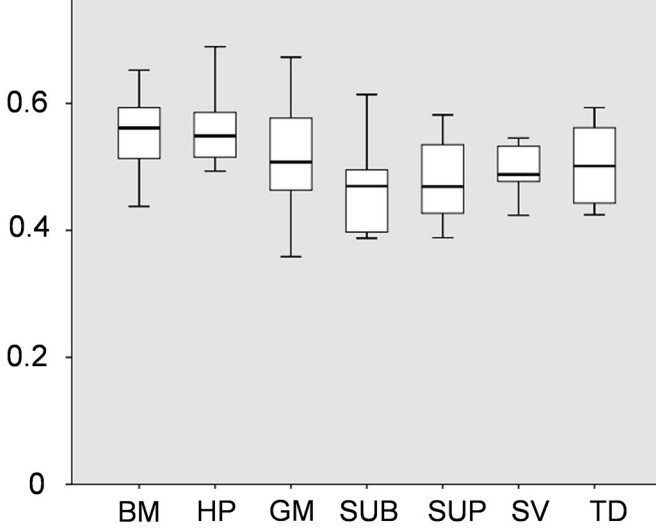
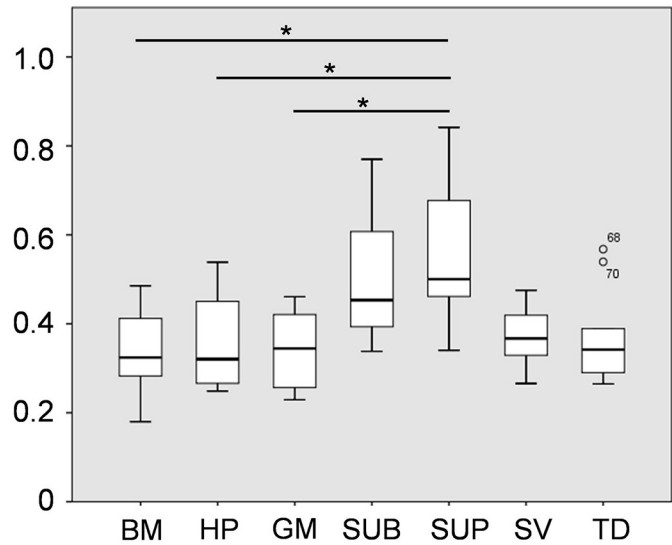

BM: buccal mucosa
HP: hard palate
GM: gingival mucosa
SUB: subgingival dental biofilm
SUP: supragingival dental biofilm
SV: saliva
TD: tongue dorsum

■ post-sleeping schedule
■ pre-sleeping schedule

**Fig 2. Alpha and beta diversity of microbiomes collected from the oral cavity.** The Chao1 index and Shannon index (A) are shown to indicate alpha diversity, and unweighted and weighted UniFrac distances between the two schedules (B) are shown to indicate intraindividual beta diversity. Asterisks indicate significant differences and circles represent outliers.

diversity was observed between the post-sleeping and pre-sleeping schedules for any oral location by unweighted UniFrac distance analysis. However, the weighted UniFrac distance of the supragingival dental biofilm between the two schedules was significantly higher than those for the buccal mucosa, hard palate, and gingival mucosa.

To investigate the variability of the microbiome collected from each oral site and in the morning versus at night, NMDS based on the Bray–Curtis distance and PERMANOVA were performed. Samples collected from different oral sites (buccal mucosa, hard palate, tongue dorsum, gingival mucosa, subgingival dental biofilm, supragingival dental biofilm, and saliva) showed statistically significant differences in composition (Fig 3, PERMANOVA, P = 0.001). The microbiome compositions of the buccal mucosa, hard palate, and gingival mucosa resembled each other. That of the tongue dorsum was similar to that of saliva.

Symbol shapes indicate the oral sampling location, and the color indicates the sample collection schedule. The number in parentheses indicates the P-value in PERMANOVA.

In addition, the post-sleep tongue dorsum microbiome was significantly different from the pre-sleep tongue dorsum microbiome (Fig 4; PERMANOVA P = 0.046). No significant difference was observed between the post-sleep and pre-sleep schedules in the microbial composition of the buccal mucosa, hard palate, gingival mucosa, subgingival dental biofilm, supragingival dental biofilm, or saliva (Fig 4; PERMANOVA: buccal mucosa, P = 0.097; hard palate, P = 0.246; gingival mucosa, P = 0.68; subgingival dental biofilm, P = 0.754; supragingival dental biofilm, P = 0.206; saliva, P = 0.081). However, there were significant differences in the composition of the overall microbiome samples from all locations and the two schedules (PERMANOVA, P = 0.007).

## Bacterial taxa at the phylum level

The bacterial composition of the biofilms at the phylum level is shown in Fig 5 and S1 Table. Biofilm-forming bacteria at oral sites belonged to five phyla: Actinobacteria, Bacteroidetes, Firmicutes, Fusobacteria, and Proteobacteria. Actinobacteria were present at lower levels on the buccal mucosa than at other sites. Firmicutes accounted for about 50% of bacteria on the buccal mucosa, hard palate, and gingival mucosa (relative abundance of Firmicutes at each site in the morning and at night, respectively: buccal mucosa, 50.7% and 60.9%; hard palate, 52.5% and 58.9%; gingival mucosa, 54.2% and 55.1%). In contrast, Firmicutes accounted for 15% to 30% of bacteria in subgingival and supragingival dental biofilms, in saliva, and on the tongue dorsum (subgingival dental biofilm, 16.3% and 14.7% in the morning and at night, respectively; supragingival dental biofilm, 19.3% and 23.5%; saliva, 29.6% and 32.0%; tongue dorsum, 27.5% and 30.3%).

## Bacterial taxa at the genus level

The bacterial composition of the biofilms at the genus level is shown in Fig 6. *Corynebacterium* and *Capnocytophaga* were relatively more abundant in the subgingival and supragingival dental biofilms than at other sites.

Genera with significant differences between morning and night are shown in Fig 7. The relative abundance of *Prevotella* on the buccal mucosa, hard palate, tongue dorsum, and in saliva was significantly higher in the morning than at night (Wilcoxon rank sum test: buccal mucosa, P = 0.013; hard palate, P = 0.028; tongue dorsum, P = 0.013; saliva, P = 0.011).

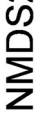

NMDS2

NMDS1

(*P*=0.007)

□ buccal mucosa
○ hard palate
△ gingival mucosa
+ subgingival dental biofilm
■ supragingival dental biofilm
● saliva
▲ tongue dorsum

**red**: post-sleeping schedule
**blue**: pre-sleeping schedule

**Fig 3. Microbial profiles of all samples.**

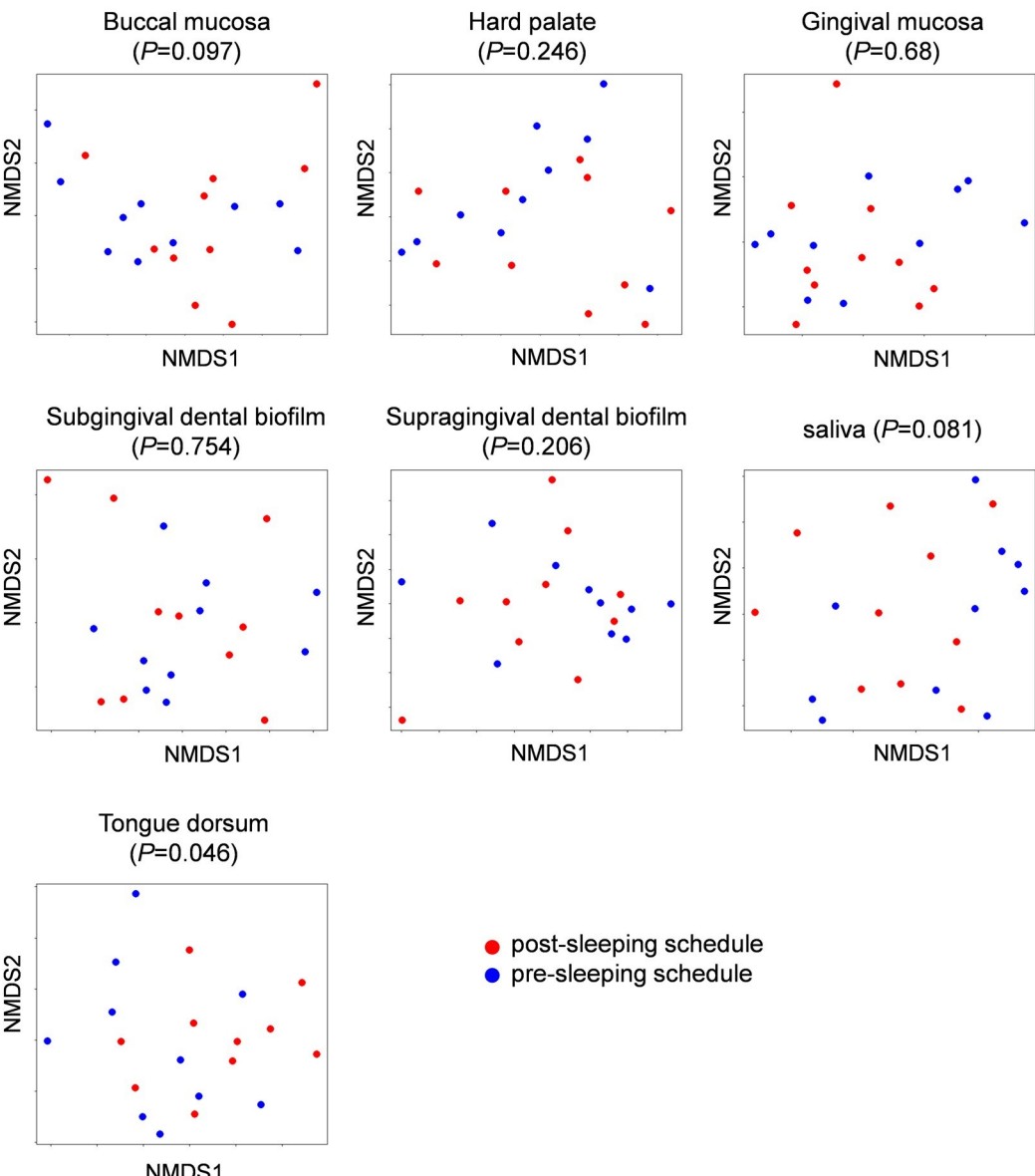

**Fig 4. Microbial profiles of samples at each location.** Non-metric multidimensional scaling and permutational multivariate analysis of variance (PERMANOVA) were performed to compare the microbial profiles at each oral location in the post-sleep and pre-sleep schedules. Data points are colored according to the sample collection schedule. Numbers in parentheses indicate the P-value determined by PERMANOVA.

*Corynebacterium* was more abundant on the buccal mucosa, hard palate, gingival mucosa, and supragingival dental biofilm in the morning than at night (Wilcoxon rank sum test: buccal mucosa, P = 0.015; hard palate, P = 0.013; gingival mucosa, P = 0.031; supragingival dental biofilm, P = 0.022). The relative abundance of *Streptococcus* on the buccal mucosa was significantly higher at night than in the morning (Wilcoxon rank sum test, P = 0.005). The relative abundance of *Rothia* on the gingival mucosa was significantly higher in the morning than at night (Wilcoxon rank sum test, P = 0.011); in contrast, the relative abundance of *Rothia* in saliva and on the tongue dorsum was significantly lower in the morning than at night (Wilcoxon rank sum test: saliva, P = 0.007; tongue dorsum, P = 0.022). The effect of sleep on the relative abundance of bacterial taxa differed depending on the oral site and bacterial taxon.

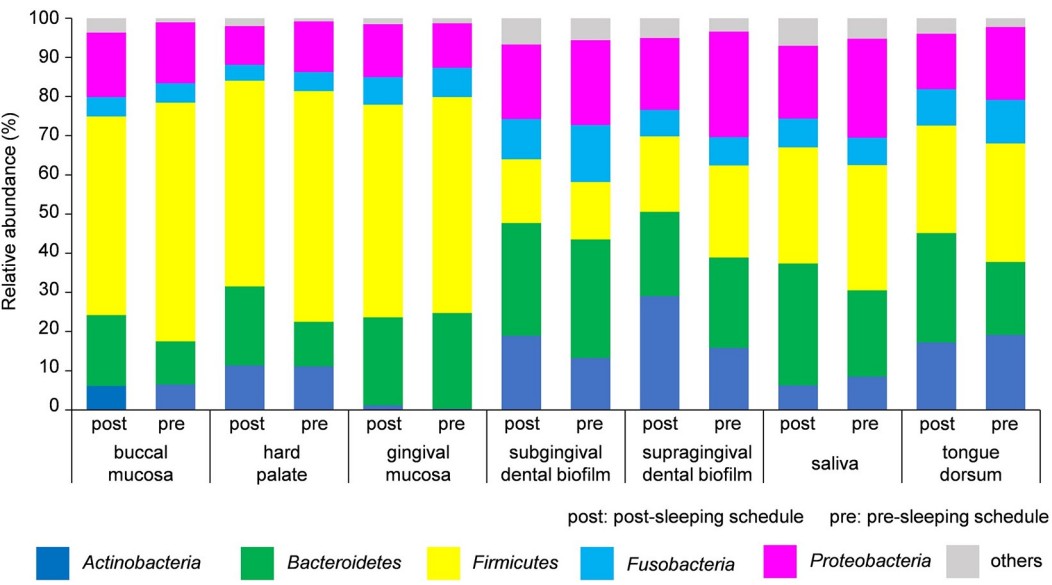

**Fig 5. Relative abundance of bacterial taxa in oral biofilms at the phylum level.** The relative abundance of bacteria at the phylum level in oral biofilms grown during the post-sleep and pre-sleep schedules. The bar colors indicate taxa.

## Discussion

This study was performed to investigate how sleep affects the microbiome diversity of oral biofilms. The effect of sleep on the microbiome of oral biofilms was investigated in detail using

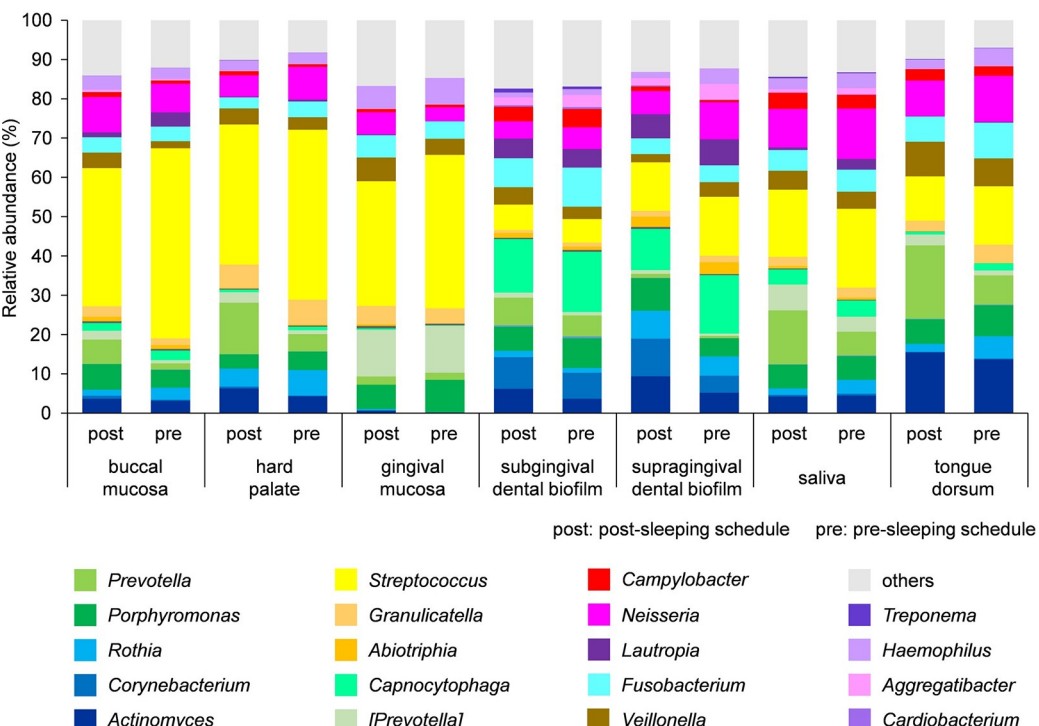

**Fig 6. Relative abundance of bacterial taxa in oral biofilms at the genus level.** The relative abundance of bacteria at the genus level in oral biofilms grown during the post-sleep and pre-sleep schedules. The colors of the bars indicate taxa.

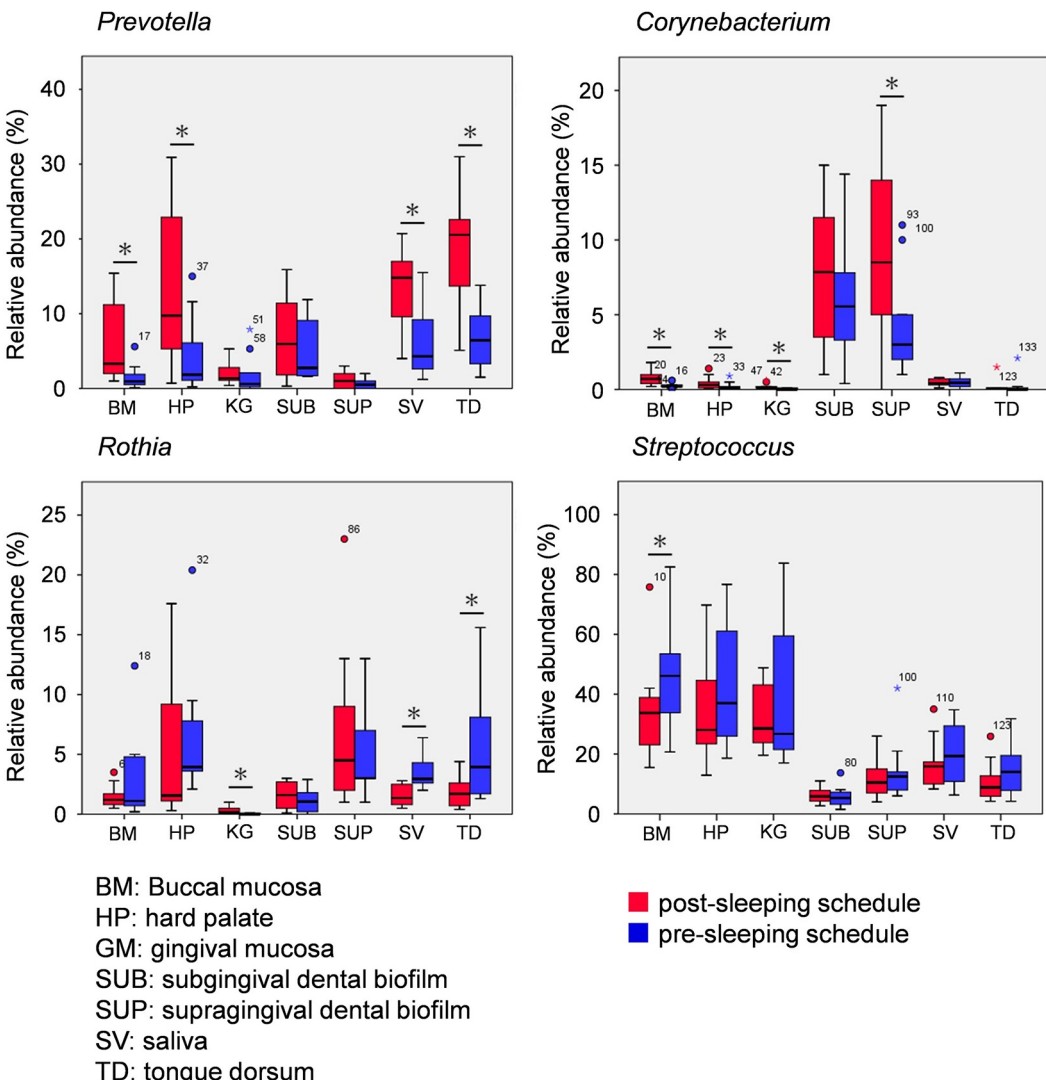

BM: Buccal mucosa
HP: hard palate
GM: gingival mucosa
SUB: subgingival dental biofilm
SUP: supragingival dental biofilm
SV: saliva
TD: tongue dorsum

■ post-sleeping schedule
■ pre-sleeping schedule

**Fig 7. Genera with significantly different relative abundances between post-sleep and pre-sleep schedules.** Significant differences were observed in four genera between post-sleep and pre-sleep schedules (Wilcoxon signed rank test, P < 0.05). Asterisks indicate significant differences and circles represent outliers.

16S rRNA gene sequence analysis. Takayasu *et al.* found circadian oscillation patterns in some genera and bacterial phenotypes of the salivary microbiome, with the relative abundance of *Prevotella* increasing between 04:00 and 12:00, that of *Gemella* and *Streptococcus* increasing between 16:00 and 00:00, that of Gram-positive species increasing between 16:00 and 04:00, and that of Gram-negative species increasing between 04:00 and 16:00 [29]. However, there is little information about circadian changes in other parts of the oral microbiome, including in the dental biofilms and tongue-coating biofilm. Therefore, we investigated microbial differences in the microbiome of oral biofilms before and after sleep.

To research the impact of sleep on biofilms, it seems reasonable to collect samples before and after sleep. However, if samples are simply taken at a series of timepoints, the biofilm age will be different when sampled before and after sleep, making it difficult to accurately assess the effects of sleep. Therefore, schedules were applied in this study to allow us to obtain samples of the same age.

It has been reported that the microbiomes of oral biofilms that form on the oral mucosa differ from those on the tooth surface, and that almost all bacteria present in oral biofilms belong to five phyla: Actinobacteria, Bacteroidetes, Firmicutes, Fusobacteria, and Proteobacteria [37, 38]. The results of the present study are consistent with those of previous reports. *Corynebacterium* and *Capnocytophaga* were predominant in the supragingival and subgingival dental biofilms, as reported previously [3]. Using Human Microbiome Project 16S rRNA gene sequencing data, Eren *et al.* reported that the microbiomes of different oral sites are distinct from each other, and, in particular, the dental biofilm is very different from that of other locations, such as saliva, buccal mucosa, tongue dorsum, and gingival mucosa [39]. Segata *et al.* investigated the bacterial composition of 10 sites in the digestive tracts of healthy subjects, including seven oral sites, and showed that the microbiomes of the 10 sites divided into four groups [40]; the microbial composition was similar between the tongue dorsum and saliva. Somineni *et al.* compared the oral microbiome in individuals with inflammatory bowel disease and healthy controls, and reported that the bacterial composition of the buccal mucosa was clearly separate from that of the tongue dorsum with or without disease [41]. Similarly, our results showed significant differences in the microbial composition at different oral sites. Samples from the tongue dorsum resembled those from saliva, but were different from those from buccal mucosa.

In this study, biofilms collected before and after sleep were compared. Regarding alpha diversity, the Chao1 index in the post-sleep schedule was significantly higher than that in the pre-sleep schedule for samples from the buccal mucosa and gingival mucosa, and the Shannon index in the post-sleep schedule was significantly higher than that in the pre-sleep schedule in samples from buccal mucosa. Some bacteria were observed only in the post-sleeping schedule, however, their relative abundance was very low; they may have been present but below the limit of detection in the pre-sleeping schedule. Another possibility is that bacteria released from other oral sites temporarily adhered to the sample sites because of a decrease in self-cleaning action at night, i.e., decrease of salivary flow, mechanical cleaning (such as eating), and mucosal movement during sleep.

Considering beta diversity, no significant difference was observed between locations in the oral cavity in unweighted UniFrac distance analysis; however, a significant difference was observed between the supragingival dental biofilm and the buccal mucosa, hard palate, and gingival mucosa in weighted UniFrac distance analysis (Fig 2B). UniFrac distance in Fig 2B shows the difference between the pre-sleeping schedule and the post-sleeping schedule. These results suggest that the microbiome composition, rather than the microbial (taxonomic) members of the various communities, are affected by sleeping. Of the seven locations in the oral cavity that we sampled, supragingival dental biofilms had the largest changes in microbiome composition before and after sleep, and are presumed to be susceptible to environmental change.

PERMANOVA showed significant differences before compared with after sleep only in the microbiota of tongue-coating biofilms. However, at the genus level, significant time-related differences were observed not only in the tongue-coating biofilm but also in the biofilm microbiomes at other locations in the oral cavity. It is possible that differences in the relative abundance of each genus between the post-sleep and pre-sleep schedule were too small to be detected with PERMANOVA. *Prevotella* and *Corynebacterium* were relatively more abundant in the post-sleep schedule than in the pre-sleep schedule in all locations in the oral cavity. A high abundance of these genera in the post-sleep schedule was observed in our previous study that used an *in situ* dental biofilm model [28]. These common tendencies among locations in the oral cavity are thought to result from environmental factors, including host immunity, saliva pH, and enzymes, which reportedly affect the oral microbiome [6]. In particular, saliva

plays a crucial role in host defense [42] and contains various antibacterial agents, such as cystatins, histatins, lactoferrin, lysozyme, mucins, statherin, and immunoglobulins [especially secretory immunoglobulin A (sIgA)]. Sarkar *et al.* explored the relationship between the salivary microbiome and cytokines [interleukin (IL)-1β, IL-6 and IL-8] in saliva in healthy subjects over 24 h. They reported that cytokine concentrations were highest at the time of waking, and the relative abundance of *Prevotella* was associated with IL-1β and IL-8 concentrations (most significantly with IL-1β) [43]. The salivary microbiome is thought to be formed by bacteria attached to other oral sites, especially the tongue dorsum [38, 39], so microbial changes on the tongue dorsum and saliva may be similar.

Several factors change in the oral environment during sleep. Concentrations of sIgA have been shown to peak during sleep [44]. The amount of glucose and protein contained in saliva during sleep is lower than that during awakening [25, 45]. The pH is high during the day (pH = 7.7) and slowly decreases during sleep (to pH 6.6). There is also a significant difference in the intraoral temperature: 33.9˚C when awake, and 35.9˚C during sleep [46]. Other factors that affect the oral microbiota may also change with periodicity. Further research is needed to clarify such interactions.

Using an *in situ* dental biofilm model, we investigated the chronological changes in an experimental dental biofilm, and revealed that the microbial composition changed with increasing bacterial counts [47]. We have also previously reported that the amount of dental biofilm was not changed pre- *versus* post-sleeping, while the dental biofilm microbiome changed [28]. In the present study, we could not estimate whether the amount of biofilm was changed by sleeping because the samples were collected by swabbing; the difference in the oral microbiome composition during sleep is considered not to be due to a change in microbial amount.

In contrast to *Prevotella* and *Corynebacterium*, which showed similar time-related changes at different oral biofilm locations, the abundance of *Rothia* fluctuated differently in different locations. The relative abundance of *Rothia* on the gingival mucosa was significantly higher on awakening, whereas levels in the saliva and on the tongue dorsum were significantly lower on awakening than before sleep. Changes in the microbiome may depend on characteristics of the surface on which the biofilm forms and on responses of the biofilm to environmental change in different locations. However, this is only inference because there remains a lack of information about environmental changes in different locations in the oral cavity during sleep. Greater knowledge about differences in the biofilm microbiome in various locations in the oral cavity and how these changes are affected by host behaviors, such as sleep, may contribute to the development of effective methods to control oral biofilms and associated diseases.

Acidic environments caused by oral bacteria lead to dental caries [8, 9]; especially on the surface of enamel, the pH drops significantly after exposure to carbohydrate [10]. The bacteria reported to produce acid in the oral cavity include members of the *Streptococcus* [48], *Veillonella* [49], and *Lactobacillus* [50]. Although *Streptococcus* and *Veillonella* were detected in this study, there was no significant difference in their levels in the dental biofilm before and after sleep.

Periodontitis is associated with obligate anaerobes in the dental biofilm [51]. In this work, a significantly higher relative abundance of *Corynebacterium* was seen in biofilms of the buccal mucosa, hard palate, gingival mucosa and in the supragingival dental biofilm on awakening compared with those before sleep. *C. matruchotii* contributes to dental biofilm mineralization, which leads to dental calculus formation [52]. Dental calculus promotes bacterial accumulation on tooth surfaces because of its rough surface, and is a risk factor for periodontitis. Ritz *et al.* examined chronological changes in the microbial composition of supragingival biofilms and showed that *Corynebacterium* increases up to 5 days, and its relative abundance is 1% after

3 days [53]. *Corynebacterium* is considered to play a central role in supragingival biofilm development; it is absent from the early microbiota and seems to bind to early colonizers [54]; late colonizers then bind to the *Corynebacterium*. Our study revealed that the relative abundance of *Corynebacterium* in the supragingival dental biofilm was significantly higher in the post-sleep schedule (9.3%) than in the pre-sleep schedule (4.2%); the same tendency was observed in the subgingival dental biofilm. Therefore, the dental biofilm may have higher potential for late-colonizing bacteria to settle on awakening than at other times of the day.

Halitosis is reportedly associated with tongue-coating biofilms [18]. In this study, *Prevotella* had higher relative abundance on the buccal mucosa, saliva, hard palate, and tongue dorsum in the post-sleep schedule than in the pre-sleep schedule. Some *Prevotella* species are related to a high concentration of methyl mercaptan [55], which is a main cause of halitosis [56]. Moreover, salivary flow and swallowing play an important role in clearance of oral bacteria and balancing the oral microbiome. [43]. Because salivary flow and swallowing decrease during sleep [25], halitosis is often observed in the morning [57]. From our findings, we consider that in addition to the low salivary flow, changes in the microbiome of the tongue dorsum and saliva may also be associated with halitosis.

There are some limitations to this study. The various sleep-related changes in the microbiomes of different locations may have resulted from differences in the characteristics of the surfaces on which the biofilms form and the responses of the biofilms to environmental changes. However, the effect of sleep on the oral environment remains unclear and is a challenge for future study. In addition, only healthy subjects without periodontitis or dental caries were investigated in this research. We found little change in the relative abundance of different genera in this study, perhaps because healthy subjects were sampled. In future study, a similar investigation into the oral microbiome of patients with oral disease will contribute to establishing effective oral care and prevention of oral disease.

In conclusion, the microbiome at different locations in the oral cavity is affected by sleep and the changes depend on the bacterial genera and the characteristics of the surface on which the oral biofilms form. The findings of this study will be useful in establishing evidence-based methods for improving oral care.

## Supporting information

**S1 Table. Relative abundance of bacterial taxa in oral biofilms at the phylum level.** Abbreviations: BM, buccal mucosa; HP, hard palate; GM, gingival mucosa; SUB, subgingival dental biofilm; SUP, supragingival dental biofilm; SV, saliva; TD, tongue dorsum; Post, post-sleeping schedule; Pre, pre-sleeping schedule.
(DOCX)

## Author Contributions

**Conceptualization:** Maki Sotozono, Nanako Kuriki, Yoko Asahi, Yuichiro Noiri, Shigeyuki Ebisu.

**Data curation:** Maki Sotozono, Nanako Kuriki, Yoko Asahi, Shota Nakamura, Mikiyo Yamaguchi.

**Formal analysis:** Daisuke Motooka.

**Funding acquisition:** Maki Sotozono, Mikako Hayashi, Shigeyuki Ebisu.

**Project administration:** Yoko Asahi.

**Supervision:** Yuichiro Noiri, Mikako Hayashi, Tetsuya Iida, Shigeyuki Ebisu.

**Validation:** Maki Sotozono, Nanako Kuriki, Yoko Asahi, Daisuke Motooka, Shota Nakamura, Tetsuya Iida, Shigeyuki Ebisu.

**Visualization:** Maki Sotozono.

**Writing – original draft:** Maki Sotozono.

**Writing – review & editing:** Nanako Kuriki, Yoko Asahi, Yuichiro Noiri, Daisuke Motooka, Shota Nakamura.

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
