## [Decision Letter · Decision Letter 0]

29 Jul 2021

PONE-D-21-11990

Impact of Sleep on the Microbiome of Oral Biofilms

PLOS ONE

Dear Dr. Asahi,

Thank you for submitting your manuscript to PLOS ONE. After careful consideration, we feel that it has merit but does not fully meet PLOS ONE’s publication criteria as it currently stands. Therefore, we invite you to submit a revised version of the manuscript that addresses the points raised during the review process.

We look forward to receiving your revised manuscript.

Kind regards,

Yiping Han, Ph.D.

Academic Editor

PLOS ONE

Journal Requirements:

2. We note that you are reporting an analysis of a microarray, next-generation sequencing, or deep sequencing data set. PLOS requires that authors comply with field-specific standards for preparation, recording, and deposition of data in repositories appropriate to their field. Please upload these data to a stable, public repository (such as ArrayExpress, Gene Expression Omnibus (GEO), DNA Data Bank of Japan (DDBJ), NCBI GenBank, NCBI Sequence Read Archive, or EMBL Nucleotide Sequence Database (ENA)). In your revised cover letter, please provide the relevant accession numbers that may be used to access these data. For a full list of recommended repositories, see http://journals.plos.org/plosone/s/data-availability#loc-omics or http://journals.plos.org/plosone/s/data-availability#loc-sequencing.

- https://www.nature.com/articles/s41598-020-80541-5

In your revision ensure you cite all your sources (including your own works), and quote or rephrase any duplicated text outside the methods section. Further consideration is dependent on these concerns being addressed.

Reviewers' comments:

Reviewer's Responses to Questions

**Comments to the Author**

1. Is the manuscript technically sound, and do the data support the conclusions?

Reviewer #1: Partly

Reviewer #2: Yes

2. Has the statistical analysis been performed appropriately and rigorously? 

Reviewer #1: Yes

Reviewer #2: Yes

3. Have the authors made all data underlying the findings in their manuscript fully available?

Reviewer #1: Yes

Reviewer #2: No

4. Is the manuscript presented in an intelligible fashion and written in standard English?

Reviewer #1: Yes

Reviewer #2: Yes

5. Review Comments to the Author

Reviewer #1: In this study titled “Impact of Sleep on the Microbiome of Oral Biofilms), the authors investigated the microbiome composition of multiple intra-oral sites in healthy subjects before and after sleep using 16s sequencing analysis. Of the 7 sites investigated, only tongue dorsum showed significant, albeit weak (P=0.046), shift in microbiome composition between pre- and post-sleep. A few genera were detected displaying significant difference in relative abundance between pre- and post-sleep at specific locations.

The reviewer likes the two-schedule setup for sample collection used in this study, which allows for more specific look at the impact of sleep on the pre-existing, mature biofilm. However, there are some issues with the experimental design, the limited information provided by the study, as well as the data interpretation that make the reviewer less enthusiastic about this study.

Main comments:

1) Experimental design: Lack of the justification for the # of subject recruited. Will the # of subjects used provide enough statistical power to allow for detection of significant change between pre- and post-sleep?

2) The new knowledge came out of this study is very limited.

Other than mainly confirming published findings, the study offers very little new knowledge. Even for a few genera detected which display significant difference in their relative abundance between pre- and post-sleep at specific locations, they were presented as “inventory lists” with no in-depth discussion/explanation as to “why” some of the difference was observed.

3) within oral microbiome, there are species level diversity with diverse physiology and pathogenesis potentials within many bacterial genera, such as Streptococci. it is worthwhile to investigate microbiome composition at species level which could potentially reveal new information related to the impact of sleep.

4) Other than microbial composition, any change in microbial “load” (absolute total abundance) pre- and post-sleep?

5) The manuscript could benefit from language editing from a native English speaker.

Other comments:

1. Page 4, line 58: change “sucrose” to “carbohydrates”

2. Page 10, line 151-153. Need more references.

3. Page 12, line 171-173. Tongue data comparison is NOT in Fig.2

4. Page 17, line 252-253. Please clarify. Assuming subject performs self-cleaning twice a day (at 8:00 and 24:00), then, biofilm collected after sleep would have less time to grow (8hrs) than biofilm collected before sleep (16hr)

5. Page 21, line 311-313. The data presented (abundance shift in one genus) is not strong enough to draw such a conclusion. Same applies to Page 12, line 321-322.

Reviewer #2: General comments:

This manuscript is focused on the investigation of the microbiome of oral biofilms affected and how it is affected by sleep. Research questions are well defined, relevant, meaningful, and original. This research fills an identified knowledge gap. The method section requires more information about the sample collection and data analyses. The results are solid (but limited) and statistical analyses are robust. The results focused on comparing the relative abundance of OTUs between different experimental schedules. The discussion should attempt explaining why the variation of the microbiome was observed between the pre-sleep and the post-sleep schedules by answering question such as “Is this variation only due to the changes in microbial abundance or due to the changes in microbial species?” An improved explanation or discussion about the relationship between sleep and environmental factors modification is warrentied.

Specific Comments:

Abstract:

L38: “changes”. Consider “changes in the microbiome composition”.

Introduction:

L44: “In that project”. It is vague. Consider using “In Human Microbiome Project or In HMP”.

L48: “microbiome construction”. Based on the context, “microbiome composition/microbial assemblages/microbial composition” is more appropriate here.

L61: Dysbiosis is not a new concept is oral biofilm

L81: “in situ”. Italic font “in situ”

Methods:

L119-120: “once 7 days before sample collection and once 24 hours before sample collection.” “once” is vague here. Consider revising “the first cleaning 7 days before sample collection and the second cleaning 24 hours before sample collection.”

L148: “The Illumina library…..”. Which Illumina kit did authors use to prepare the library? What is the sequencing method? Single or paired end sequencing? The length of sequencing? (150bp, 250bp or 300bp), the total number of reads

L151-152: “operative taxonomic units”. Consider revising to “operative taxonomic units (OTUs)”

L151-152: The version of the Green Genes database is missing.

L152-153 : The version of software QIIME is missing. It is not clear if the authors checked the chimeras or remove any singletons. When authors analyze the data, it is not clear whether they rarefy the OTU table.

L157: The version of R software is needed.

At the end of the Materials and Methods, authors need an separate paragraph, which gives the information of NCBI SRA or similar database submission.

Results

It is suggested that the authors also examining the microbial diversity change between different experiment schedules or different oral locations. This can be done by calculating phylogenetic diversity in QIIME. By doing this, we will know whether the microbial composition is significantly different across experimental schedules. For example, are there any bacteria only found in the pre-sleep schedule but not in the post-sleep schedule. It would be interesting to investigate the diversity related to phylogenetics. I suggest authors checking phylogenetic diversity (e.g., UniFrac).

Figure 2. Panel B does not bring any information. Square and triangles instead of removing the color coding of the body sites would be beneficial

L164: Authors only reported beta diversity. It is not clear about the alpha diversity. For example, how many OTUs or species were found in the pre-sleep schedule and in the post-sleep schedule, respectively.

L180: “microbiome”. Consider revising to “microbial composition or microbiome composition”.

L194-195: “five phyla: Actinobacteria, Bacteroidetes, Firmicutes, Fusobacteria, and Proteobacteria. Actinobacteria were present………”. The relative abundance (%) of each phylum could be reported in supplemental data (only the phylum Firmicutes has been reported so far).

Discussion

The authors reference the HMP work but fail to do a comparison to articles were multisite analysis were performed [Segata et al Genome Biol 2012] [Eren PNAS 2014] [Somineni Infl Bowel Dis 2021]

L240: “affects the microbiome”. It should be more specific here. For example, “affects the microbiome abundance”, “affect the microbiome diversity”, etc.

L274-276: Please see my comments above. Can authors do a further discussion about whether environmental factors would contribute to the changes in the microbiome in this study?

L321-322: No supporting data for this statement.

6. PLOS authors have the option to publish the peer review history of their article (what does this mean?). If published, this will include your full peer review and any attached files.

Reviewer #1: No

Reviewer #2: No

---

## [Author Response · Author response to Decision Letter 0]

8 Sep 2021

Responses to Academic Editor

We appreciate the time and effort you have dedicated to providing insightful feedback on ways to strengthen our paper.

Response: We have checked PLOS ONE’s style requirements, and adapted our manuscript to them.

2. We note that you are reporting an analysis of a microarray, next-generation sequencing, or deep sequencing data set. PLOS requires that authors comply with field-specific standards for preparation, recording, and deposition of data in repositories appropriate to their field. Please upload these data to a stable, public repository (such as ArrayExpress, Gene Expression Omnibus (GEO), DNA Data Bank of Japan (DDBJ), NCBI GenBank, NCBI Sequence Read Archive, or EMBL Nucleotide Sequence Database (ENA)). In your revised cover letter, please provide the relevant accession numbers that may be used to access these data. For a full list of recommended repositories, see http://journals.plos.org/plosone/s/data-availability#loc-omics or http://journals.plos.org/plosone/s/data-availability#loc-sequencing.

Response: The 16S rRNA gene sequencing data have been uploaded to the DNA Data Bank of Japan (DDBJ), and the accession number (DRA011991) has been included in our cover letter and the revised manuscript (page 9, lines 140–141).

- https://www.nature.com/articles/s41598-020-80541-5

In your revision ensure you cite all your sources (including your own works), and quote or rephrase any duplicated text outside the methods section. Further consideration is dependent on these concerns being addressed.

Response: We have rephrased text that was duplicated from our previous publication, except in the Methods section (indicated by blue highlights in the revised manuscript).

Responses to Reviewer #1

We greatly appreciate the Reviewer’s insightful comments, which have aided us in significantly improving our paper.

Responses to main comments:

1. Experimental design: Lack of the justification for the # of subject recruited. Will the # of subjects used provide enough statistical power to allow for detection of significant change between pre- and post-sleep?

Response: Although there was some variation among individuals, a similar tendency was observed among all 10 of our participants. Additionally, the sample size in similar dental biofilm studies was the same as the sample size in our study (e.g., Sci. Rep. 2021; 11: 138. doi: 10.1038/s41598-020-80541-5, npj Biofilms Microbiomes 2016; 10: 2:16018, Sci. Rep. 2015; 5: 8136). Thus, while a larger sample size would be better, we believe that the sample size of our study is sufficient to support our conclusions.

2. The new knowledge came out of this study is very limited.

Other than mainly confirming published findings, the study offers very little new knowledge. Even for a few genera detected which display significant difference in their relative abundance between pre- and post-sleep at specific locations, they were presented as “inventory lists” with no in-depth discussion/explanation as to “why” some of the difference was observed.

Response: This study did not compare healthy and diseased subjects, but was a study of diurnal variation in healthy subjects, so the difference in microbiota was expected to be small. Though the changes in the oral environment during sleep and the association between the oral bacteria and the oral environment have not yet been fully elucidated, we have added the following discussion:

“Sarkar et al. explored the relationship between the salivary microbiome and cytokines [interleukin (IL)-1β, IL-6 and IL-8] in saliva in healthy subjects over 24 h. They reported that cytokine concentrations were highest at the time of waking, and the relative abundance of Prevotella was associated with IL-1β and IL-8 concentrations (most significantly with IL-1β) [44]. The salivary microbiome is thought to be formed by bacteria attached to other oral sites, especially the tongue dorsum [38, 39], so microbial changes on the tongue dorsum and saliva may be similar. 

Several factors change in the oral environment during sleep. Concentrations of sIgA have been shown to peak during sleep [45]. The amount of glucose and protein contained in saliva during sleep is lower than that during awakening [46, 47]. The pH is high during the day (pH = 7.7) and slowly decreases during sleep (to pH 6.6). There is also a significant difference in the intraoral temperature: 33.9 °C when awake, and 35.9 °C during sleep [48]. Other factors that affect the oral microbiota may also change with periodicity. Further research is needed to clarify such interactions.” (page20-21, line319-332)

3. Within oral microbiome, there are species level diversity with diverse physiology and pathogenesis potentials within many bacterial genera, such as Streptococci. It is worthwhile to investigate microbiome composition at species level which could potentially reveal new information related to the impact of sleep.

Response: The sequencing in this study was conducted using the MiSeq system, from which we obtained sequences of 16S rRNA gene V1–V2 regions (about 300 bp). As the Reviewer points out, it would be valuable to investigate microbial composition at the species level to enable more detailed consideration. However, species-level analysis is difficult (having low classification rate and accuracy) using such partial 16S rRNA gene sequences. Full-length 16S rRNA gene sequencing, which would enable species-level analysis, is possible using PacBio or Oxford Nanopore technology, but the sequence error rates are higher than in MiSeq. Shotgun metagenomic analysis is another method of analysis at the species level, but it is time-consuming and expensive. Based on the data we obtained here, we chose the more reliable genus level for analysis. Nevertheless, analysis at the species level is a prospect for future studies.

4. Other than microbial composition, any change in microbial “load” (absolute total abundance) pre- and post-sleep?

Response: In this study, oral biofilm samples from various places in the oral cavity were collected by swabbing. So, it is difficult to estimate the amount of biofilm per unit. However, using an in situ dental biofilm model, which was quantitatively evaluated, we previously reported that the amount of dental biofilm (the number of bacteria and the biovolume) was not affected by sleep (Sci. Rep. 2021; 11: 138). We added the following text in the Discussion of the revised manuscript:

“Using an in situ dental biofilm model, we investigated the chronological changes in an experimental dental biofilm, and revealed that the microbial composition changed with increasing bacterial counts [49]. We have also previously reported that the amount of dental biofilm was not changed pre- versus post-sleeping, while the dental biofilm microbiome changed [28]. In the present study, we could not estimate whether the amount of biofilm was changed by sleeping because the samples were collected by swabbing.” (page 21, line 333-338)

5. The manuscript could benefit from language editing from a native English speaker.

Response: We have had the English and grammar in the manuscript checked by a native English speaker from a professional editing company.

Responses to other comments:

1. Page 4, line 58: change “sucrose” to “carbohydrates”

Response: In accordance with the Reviewer’s suggestion, we have changed “sucrose” to “carbohydrates” in the revised manuscript (page 3, line 48).

2. Page 10, line 151-153. Need more references.

Responses: We have added references, as follows:

“The sequences were processed and clustered into operational taxonomic units (OTUs) with a 97% similarity cutoff by using the Greengenes database (v. 13.8) [35]. The results of sequences were analyzed by using the Quantitative Insights into Microbial Ecology pipeline (v. 1.9.1) [36].” (page 9, line 138-139)

“[35] DeSantis TZ, Hugenholtz P, Larsen N, Rojas M, Brodie EL, Keller K, et al. Greengenes, a chimera-checked 16S rRNA gene database and workbench compatible with ARB. Appl Environ Microbiol. 2006; 72: 5069–5072.

[36] Caporaso JG, Kuczynski J, Stombaugh J, Bittinger K, Bushman FD, Costello EK, et al. QIIME allows analysis of high-throughput community sequencing data. Nat Methods. 2010; 7: 335–336.” (page 63, line 476-481)

3. Page 12, line 171-173. Tongue data comparison is NOT in Fig.2.

Responses: “Tongue” in line 171–173 meant the same as “tongue dorsum” in Fig. 2. The words “tongue” and “tongue dorsum” that were used to mean the sample taken from the tongue were mixed in the manuscript and the figure, and that caused confusion, for which we apologize. The expression for the tongue sample has been unified to “tongue dorsum” throughout the revised manuscript (page 8, line 121; page 12, lines 175, 185-186; page 14, lines 211, 213; page 15, lines 229, 231, 239, 241; page 18, lines 275; page 22, lines 342; page 24, line 375).

4. Page 17, line 252-253. Please clarify. Assuming subject performs self-cleaning twice a day (at 8:00 and 24:00), then, biofilm collected after sleep would have less time to grow (8hrs) than biofilm collected before sleep (16hr).

Responses: As described in the Methods, in this study, the subjects received professional oral care twice, 7 d and 24 h before sampling. After professional care 24 h before sampling, they avoided oral self-care until after sample collection. For example, in the post-sleeping schedule, the subjects had their mouths cleaned at 08:00 and they avoided tooth brushing until sample collection (08:00 the next day). In the post-sleeping schedule, they had their mouths cleaned at 00:00 and avoided oral self-care until sample collection, which was performed the next day at 00:00. We have added the following text to the Discussion: 

“However, if samples are simply taken at a series of timepoints, the biofilm age will be different when sampled before and after sleep, making it difficult to accurately assess the effects of sleep.” (page 17, line 262-264)

5. Page 21, line 311-313. The data presented (abundance shift in one genus) is not strong enough to draw such a conclusion. Same applies to Page 12, line 321-322.

Responses: As the Reviewer points out, it was an overestimate to discuss that a relative abundance shift in one genus may lead to higher periodontal disease-related pathogenicity. However, recent structural analysis reports that Corynebacterium is the cornerstone of dental biofilm development. Corynebacterium seems to bind to the early colonizers, and then late colonizers bind to Corynebacterium. Therefore, we have revised the manuscript as follows:

“Corynebacterium is considered to play a central role in supragingival biofilm development; it is absent from the early microbiota and seems to bind to early colonizers [56]; late colonizers then bind to the Corynebacterium.…Therefore, the dental biofilm may have higher potential for late-colonizing bacteria to settle on awakening than at other times of the day.” (page21, line 365-368, 371-372)

Additionally, we overstated the relationship with halitosis. We have revised the manuscript as follows:

“From our findings, we consider that in addition to the low salivary flow, changes in the microbiome of the tongue dorsum and saliva may also be associated with halitosis.” (page 24, line 380-381)

Responses to Reviewer #2

We thank the Reviewer for the very helpful comments, which have aided us in significantly improving our paper.

Responses to General Comments:

This manuscript is focused on the investigation of the microbiome of oral biofilms affected and how it is affected by sleep. Research questions are well defined, relevant, meaningful, and original. This research fills an identified knowledge gap. The method section requires more information about the sample collection and data analyses. The results are solid (but limited) and statistical analyses are robust. The results focused on comparing the relative abundance of OTUs between different experimental schedules. The discussion should attempt explaining why the variation of the microbiome was observed between the pre-sleep and the post-sleep schedules by answering question such as “Is this variation only due to the changes in microbial abundance or due to the changes in microbial species?” An improved explanation or discussion about the relationship between sleep and environmental factors modification is warrentied.

Responses: We have improved the Discussion with reference to the Reviewer’s question.

Responses to Specific Comments:

Abstract:

1. L38: “changes”. Consider “changes in the microbiome composition”.

Responses: In accordance with Reviewer’s suggestion, we have changed “changes” to “changes in the microbiome composition” (page 2, line 30).

Introduction:

2. L44: “In that project”. It is vague. Consider using “In Human Microbiome Project or In HMP”.

Responses: We have changed “In that project” to “In the Human Microbiome Project” (page 3, line 35-36).

3. L48: “microbiome construction”. Based on the context, “microbiome composition/microbial assemblages/microbial composition” is more appropriate here.

Responses: We have changed “microbiome construction” to “microbiome composition” (page 3, line 40).

4. L61: Dysbiosis is not a new concept is oral biofilm.

Responses: We have removed “recently” from the revised manuscript (page 4, line 50-51).

5. L81: “in situ”. Italic font “in situ”.

Responses: We have put “in situ” in italics (page 5, line 68).

Methods:

6. L119-120: “once 7 days before sample collection and once 24 hours before sample collection.” “once” is vague here. Consider revising “the first cleaning 7 days before sample collection and the second cleaning 24 hours before sample collection.”

Responses: Thank you for this helpful suggestion. We have revised “once 7 days before sample collection and once 24 hours before sample collection” to “the first cleaning 7 days before sample collection and the second cleaning 24 h before sample collection” (page 7, line 104-105).

7. L148: “The Illumina library…..”. Which Illumina kit did authors use to prepare the library? What is the sequencing method? Single or paired end sequencing? The length of sequencing? (150bp, 250bp or 300bp), the total number of reads.

Responses: The Illumina kit used was the MiSeq Reagent Kit v2 (500 cycles) and we sequenced in a 251-bp paired-end run. We have added the sentence below:

“Sequencing (251-bp paired-end) was performed using MiSeq Reagent Kit v2 (500 cycles) and a MiSeq instrument (Illumina Inc.)” (page9, line 135)

The total number of reads was 10,169,418. We have revised the manuscript as follows:

“The total number of reads was 10,169, 418, and the average read count was 72,639.” (page 11, line 154-155)

8. L151-152: “operative taxonomic units”. Consider revising to “operative taxonomic units (OTUs)”.

Responses: We have added “(OTUs)” just after “operative taxonomic units” (page 9, line 137).

9. L151-152: The version of the Green Genes database is missing.

Responses: We used Greengenes database ver. 13.8; this has been added to the revised manuscript (page 9, line 138).

10. L152-153: The version of software QIIME is missing. It is not clear if the authors checked the chimeras or remove any singletons. When authors analyze the data, it is not clear whether they rarefy the OTU table.

Responses: The version of QIIME used (v. 1.9.1) has been added to the revised manuscript (page 9, line 139).

Chimera sequences and singletons were not removed. We checked the alpha rarefaction, and analyzed the data at 20,000 reads.

11. L157: The version of R software is needed.

Responses: The version of R software used (v. 3.6.1) has been added to the revised manuscript (page 10, line 145).

12. At the end of the Materials and Methods, authors need a separate paragraph, which gives the information of NCBI SRA or similar database submission.

Responses: The 16S rRNA sequencing data were deposited in the DNA Data Bank of Japan (DDBJ) under accession number DRA011991. We have added the following sentence:

“The 16S rRNA amplicon sequencing data from this study was deposited in the DNA Data Bank of Japan (DDBJ) with accession number DRA011991.” (page 9, line 140-141)

Results:

13. It is suggested that the authors also examining the microbial diversity change between different experiment schedules or different oral locations. This can be done by calculating phylogenetic diversity in QIIME. By doing this, we will know whether the microbial composition is significantly different across experimental schedules. For example, are there any bacteria only found in the pre-sleep schedule but not in the post-sleep schedule. It would be interesting to investigate the diversity related to phylogenetics. I suggest authors checking phylogenetic diversity (e.g., UniFrac).

Responses: Thank you for this helpful suggestion. Unweighted and weighted UniFrac distance was checked as beta diversity. The results and discussion were added to the revised manuscript: 

“No significant difference in intraindividual diversity was observed between the post-sleeping and pre-sleeping schedules for any oral location by unweighted UniFrac distance analysis. However, the weighted UniFrac distance of the supragingival dental biofilm between the two schedules was significantly higher than those for the buccal mucosa, hard palate, and gingival mucosa.” (page11, line 159-164)

“Regarding alpha diversity, the Chao1 index in the post-sleep schedule was significantly higher than that in the pre-sleep schedule for samples from the buccal mucosa and gingival mucosa, and the Shannon index in the post-sleep schedule was significantly higher than that in the pre-sleep schedule in samples from buccal mucosa. Some bacteria were observed only in the post-sleeping schedule, however, their relative abundance was very low; they may have been present but below the limit of detection in the pre-sleeping schedule. Another possibility is that bacteria released from other oral sites temporarily adhered to the sample sites because of a decrease in self-cleaning action at night, i.e., decrease of salivary flow, mechanical cleaning (such as eating), and mucosal movement during sleep.

Considering beta diversity, no significant difference was observed between locations in the oral cavity in unweighted UniFrac distance analysis; however, a significant difference was observed between the supragingival dental biofilm and the buccal mucosa, hard palate, and gingival mucosa in weighted UniFrac distance analysis (Fig. 2B). UniFrac distance in Fig. 2B shows the difference between the pre-sleeping schedule and the post-sleeping schedule. These results suggest that the microbiome composition, rather than the microbial (taxonomic) members of the various communities, are affected by sleeping. Of the seven locations in the oral cavity that we sampled, supragingival dental biofilms had the largest changes in microbiome composition before and after sleep, and are presumed to be susceptible to environmental change.” (page19-20, line285-304)

14. Figure 2. Panel B does not bring any information. Square and triangles instead of removing the color coding of the body sites would be beneficial.

Responses: In accordance with Reviewer’s helpful suggestion, we have changed the Figure using both colors and shapes. To match the following figures (post-sleeping schedule, red; pre-sleeping schedule, blue), the schedule is shown in colors and the sample sites are shown using shapes. 

15. L164: Authors only reported beta diversity. It is not clear about the alpha diversity. For example, how many OTUs or species were found in the pre-sleep schedule and in the post-sleep schedule, respectively.

Responses: Data on alpha diversity were investigated. The results were added to the revised manuscript (page 12, line 171–177, Fig 2A):

“Alpha diversity (Chao1 and Shannon indexes) are shown in Fig 2A, and beta diversity (UniFrac distances) in Fig 2B. There was a significant difference in the Chao1 index between the post-sleeping and pre-sleeping schedules at the buccal mucosa and gingival mucosa (buccal mucosa P = 0.022, gingival mucosa P = 0.037). There was also a significant difference in the Shannon index at the buccal mucosa (P = 0.007).” (page 11, line 155-159)

There was also a significant difference in the species between the post-sleeping and pre-sleeping schedules at the buccal mucosa and gingival mucosa.

16: L180: “microbiome”. Consider revising to “microbial composition or microbiome composition”.

Responses: We revised “microbiome” to “microbial composition” (page 13, line 188).

17. L194-195: “five phyla: Actinobacteria, Bacteroidetes, Firmicutes, Fusobacteria, and Proteobacteria. Actinobacteria were present………”. The relative abundance (%) of each phylum could be reported in supplemental data (only the phylum Firmicutes has been reported so far).

Responses: Per the Reviewer’s suggestion, we have added the relative abundance (%) of each of the five phyla as supplemental data (Supporting Information, S1 Table).

Discussion:

18. The authors reference the HMP work but fail to do a comparison to articles were multisite analysis were performed [Segata et al Genome Biol 2012] [Eren PNAS 2014] [Somineni Infl Bowel Dis 2021].

Responses: We have compared our data with the articles that were suggested by the Reviewer, and added discussion to the revised manuscript:

“Using Human Microbiome Project 16S rRNA gene sequencing data, Eren et al. reported that the microbiomes of different oral sites are distinct from each other, and, in particular, the dental biofilm is very different from that of other locations, such as saliva, buccal mucosa, tongue dorsum, and gingival mucosa [39]. Segata et al. investigated the bacterial composition of 10 sites in the digestive tracts of healthy subjects, including seven oral sites, and showed that the microbiomes of the 10 sites divided into four groups [40]; the microbial composition was similar between the tongue dorsum and saliva. Somineni et al. compared the oral microbiome in individuals with inflammatory bowel disease and healthy controls, and reported that the bacterial composition of the buccal mucosa was clearly separate from that of the tongue dorsum with or without disease [41]. Similarly, our results showed significant differences in the microbial composition at different oral sites. Samples from the tongue dorsum resembled those from saliva, but were different from those from buccal mucosa.” (page 18, line 271-284)

19. L240: “affects the microbiome”. It should be more specific here. For example, “affects the microbiome abundance”, “affect the microbiome diversity”, etc.

Responses: In accordance with the Reviewer’s suggestion, we revised “affects the microbiome” to “affects the microbiome diversity” (page 17, line 250).

20. L274-276: Please see my comments above. Can authors do a further discussion about whether environmental factors would contribute to the changes in the microbiome in this study?

Responses: How sleep changes the oral environment, and the relationship between changes in the oral environment and the oral microbiome, have not yet been fully elucidated. However, we considered as much as possible and added the following text to the Discussion:

“Sarkar et al. explored the relationship between the salivary microbiome and cytokines [interleukin (IL)-1β, IL-6 and IL-8] in saliva in healthy subjects over 24 h. They reported that cytokine concentrations were highest at the time of waking, and the relative abundance of Prevotella was associated with IL-1β and IL-8 concentrations (most significantly with IL-1β) [44]. The salivary microbiome is thought to be formed by bacteria attached to other oral sites, especially the tongue dorsum [38, 39], so microbial changes on the tongue dorsum and saliva may be similar. 

Several factors change in the oral environment during sleep. Concentrations of sIgA have been shown to peak during sleep [45]. The amount of glucose and protein contained in saliva during sleep is lower than that during awakening [46, 47]. The pH is high during the day (pH = 7.7) and slowly decreases during sleep (to pH 6.6). There is also a significant difference in the intraoral temperature: 33.9 °C when awake, and 35.9 °C during sleep [48]. Other factors that affect the oral microbiota may also change with periodicity. Further research is needed to clarify such interactions.” (page20-21, line319-332)

21. L321-322: No supporting data for this statement.

Responses: As the Reviewer suggested, the discussion about halitosis was exaggerated, we have revised the manuscript, as follows:

“From our findings, we consider that in addition to the low salivary flow, changes in the microbiome of the tongue dorsum and saliva may also be associated with halitosis” (page 24, line 380-381)

---

## [Decision Letter · Decision Letter 1]

28 Oct 2021

Impact of sleep on the microbiome of oral biofilms

PONE-D-21-11990R1

Dear Dr. Asahi,

We’re pleased to inform you that your manuscript has been judged scientifically suitable for publication and will be formally accepted for publication once it meets all outstanding technical requirements.

Kind regards,

Yiping Han, Ph.D.

Academic Editor

PLOS ONE

Additional Editor Comments (optional):

Reviewers' comments:

Reviewer's Responses to Questions

**Comments to the Author**

1. If the authors have adequately addressed your comments raised in a previous round of review and you feel that this manuscript is now acceptable for publication, you may indicate that here to bypass the “Comments to the Author” section, enter your conflict of interest statement in the “Confidential to Editor” section, and submit your "Accept" recommendation.

Reviewer #1: All comments have been addressed

2. Is the manuscript technically sound, and do the data support the conclusions?

Reviewer #1: Yes

3. Has the statistical analysis been performed appropriately and rigorously? 

Reviewer #1: Yes

4. Have the authors made all data underlying the findings in their manuscript fully available?

Reviewer #1: Yes

5. Is the manuscript presented in an intelligible fashion and written in standard English?

Reviewer #1: Yes

6. Review Comments to the Author

Reviewer #1: the authors have adequately addressed the reviewer's concerns. The reviewer recommends accepting the manuscript.

7. PLOS authors have the option to publish the peer review history of their article (what does this mean?). If published, this will include your full peer review and any attached files.

Reviewer #1: **Yes: **Xuesong He

---

## [Editor Report · Acceptance letter]

1 Dec 2021

PONE-D-21-11990R1 

Impact of sleep on the microbiome of oral biofilms 

Dear Dr. Asahi:

I'm pleased to inform you that your manuscript has been deemed suitable for publication in PLOS ONE. Congratulations! Your manuscript is now with our production department. 

Kind regards, 

on behalf of

Dr. Yiping Han 

Academic Editor

PLOS ONE